# Effects of Fermentation on Standardized Ileal Digestibility of Amino Acids and Apparent Metabolizable Energy in Rapeseed Meal Fed to Broiler Chickens

**DOI:** 10.3390/ani10101774

**Published:** 2020-10-01

**Authors:** Zhengke Wu, Jiao Liu, Jiang Chen, Shoaib Ahmed Pirzado, Yang Li, Huiyi Cai, Guohua Liu

**Affiliations:** 1Key Laboratory of Feed Biotechnology of Agricultural Ministry and Rural Affairs, National Engineering Research Center of Biological Feed, Feed Research Institute of Chinese Academy of Agricultural Science, Beijing 100081, China; wzk199107@163.com (Z.W.); liujiao_0214@163.com (J.L.); jiangchen363@163.com (J.C.); dr.pirzado@gmail.com (S.A.P.); liyang8906@163.com (Y.L.); caihuiyi@caas.cn (H.C.); 2Department of Animal Nutrition, Sindh Agriculture University, Tnadojam 70060, Pakistan

**Keywords:** fermentation, nutritive value, apparent metabolizable energy, standardized ileal digestibility of amino acids, broilers

## Abstract

**Simple Summary:**

Rapeseed meal (RSM) is a by-product of rapeseed oil production. Owing to its lower cost and abundant sulfur-containing amino acids, RSM can be used for replacing soybean meal in broiler diets. However, its use is limited by the presence of numerous anti-nutritional factors. As an ancient technique to convert the complex substrates into simple compounds by a number of microorganisms, microbial solid-state fermentation (SSF) has been shown as an effective way to eliminate or reduce anti-nutritional factors in RSM and improve growth performance when fed to animals. This improvement is not yet clear; in particular, the understanding of the feeding nutritional value of fermented rapeseed meal (FRSM) is not very well studied. Hence, the trial is conducted to investigate the effects of fermentation on standardized ileal digestibility (SID) of amino acids and apparent metabolizable energy (AME) in RSM fed to broiler chickens. According to our findings, fermentation had a significant effect on the chemical composition of RSM. In comparison to RSM, FRSM had greater nitrogen-corrected apparent metabolizable energy (AMEn) values and SID of amino acids. FRSM was nutritionally superior to RSM for use in broiler diets.

**Abstract:**

Rapeseed meal (RSM) is a common protein ingredient in animal diets, while the proportion of RSM in diets is limited because of its anti-nutritional factors. Fermentation based on mixed microbial strains appears to be a suitable approach to improve the nutritive value of rapeseed meal in animal feed. In this study, we evaluated the effects of fermentation on the apparent metabolizable energy (AME) values and standardized ileal digestibility (SID) of amino acids in RSM fed broilers. The AME and nitrogen-corrected apparent metabolizable energy (AMEn) values of RSM and fermented rapeseed meal (FRSM) were determined by the substitution method, with RSM and FRSM proportionally replacing the energy-yielding components of the basal diet by 30%. Results show that fermentation improved AME and AMEn of RSM from 7.44 to 8.51 MJ/kg and from 7.17 to 8.26 MJ/kg, respectively. In the second experiment, two experimental diets were formulated, with RSM and FRSM being the sole sources of amino acids. A nitrogen-free diet (NFD) was also formulated to determine endogenous amino acids losses (EAAL). Feeding on FRSM resulted in higher (*p* < 0.05) apparent ileal digestibility (AID) and SID of alanine, valine, isoleucine, leucine, tyrosine, lysine, arginine, and phenylalanine. No significant differences between RSM and FRSM were found for AID and SID of asparagine, histidine, threonine, serine, glutamine, praline, glycine, methionine, and cystine. FRSM had greater AMEn values and SID of amino acids compared to RSM, therefore, FRSM was nutritionally superior to RSM in broiler diets.

## 1. Introduction

Rapeseed meal (RSM) is a by-product of rapeseed oil production, containing 35–40% crude protein (CP) with abundant sulfur-containing amino acids, and a potential substitute for soybean meal in broiler diets [1,2,3]. However, the amount of RSM that can be added to broiler diets is limited because of its anti-nutritional factors, such as glucosinolates, erucic acid, phytic acid, tannins, and non-starch polysaccharides [4]. Glucosinolates and its secondary metabolites are considered as toxic agents that affect growth performance and health status of animals, and, thus, no more than 10% of rapeseed meal (glucosinolates: <25 μmol/g) is recommend in broiler diets [5,6]. In order to use RSM efficiently, various processing techniques are employed to lower the levels of anti-nutritional factors in RSM, including physical, chemical, enzyme hydrolysis, and biological pretreatments [4,7]. Most of these methods have drawbacks, such as loss of proteins, high cost, reagent residues, and commercial infeasibility. Hydrothermal treatment can break weak bonds between polysaccharides, which contribute a lot to the degradation of glucosinolates, because glucosinolates are a sugar derivative, but high temperatures may increase protein and free amino acid damage. In addition, high temperatures may decrease protein digestibility and lower the nutritive values of RSM [8,9].

Fermentation is an ancient technique to convert the complex substrates into simple compounds by a number of microorganisms, and it is widely used in food processing and pharmaceutical industries [10,11]. In recent years, many advantages have been claimed for fermented rapeseed meal, including increased availability of protein and energy, destruction of anti-nutritive factors, improved broiler gastrointestinal tract microecology, and health and production performance; hence, leading to a wider choice of rapeseed meal that can be employed in feed formulations [12,13]. According to Chiang et al. (2010) [14], the pH values decreased by 9%, and the population of *Lactobacilli fermentum* increased 58% when rapeseed meal was fermented by mixed strains of *Lactobacillus fermentum, Bacillus subtilis, Saccharomyces cerevisiae,* and *Enterococcus calcium*. Fermented rapeseed meal (FRSM) has a positive effect on growth performance when fed to animals. In a study by Drazbo et al. (2018) [4], RSM fermented by commercial 6-phytase enzyme, expressed in *Pichia pastoris*, had reduced levels of glucosinolates and phytate-phosphorus, and the use of FRSM in diets increased the final body weight of turkeys. Ashayerizadeh et al. (2018) [12] stated that replacement of soybean meal with FRSM improved the growth performance as well as antioxidant capacity and meat quality of broiler chickens. Chiang et al. [14] also reported that the growth performance and intestinal morphology were improved when broilers were fed with fermented RSM. Due to the low levels of anti-nutrients in fermented RSM, it is possible that it will be feasible to increase the levels in broiler diets (where this is available). This option needs to be carefully investigated, especially before the nutritional value is fully evaluated.

These studies suggest that greater quantities of fermented rapeseed meal (FRSM) can be used in broiler diets to reduce the cost of broiler production. To study the potential for greater use of FRSM in broiler diets, a better understanding of the nutritional value of FRSM is necessary. Based on previous research about FRSM, we hypothesized that FRSM had greater AMEn values and standardized ileal digestibility (SID) of amino acids in comparison to RSM. 

## 2. Material and Methods

All animal management and experimental procedures for this study were approved by the Animal Ethics Committee of the Chinese Academy of Agricultural Sciences, and performed according to the guidelines for animal experiments set by the National Institute of Animal Health (Statement no. AEC-CAAS-20181208). The authors confirm that they have followed European Union (EU) standards for the protection of animals used for scientific purposes [15].

### 2.1. Conventional Rapeseed Meal and Fermented Rapeseed Meal

The RSM and FRSM (Table 1) samples used in this study were collected from the same variety of RSM, harvested in June 2016, and collected in April 2017 for testing, from Jiangxi Province, China. The FRSM was fermented by The Key Laboratory of Feed Biotechnology of Agricultural Ministry, Feed Research Institute of Chinese Academy of Agricultural Sciences. The fermentation conditions were optimized by previous lab research (Wu et al. In Chinese) [16].

### 2.2. Fermented Rapeseed Meal Processing Conditions

Strains of *Lactobacillus acidophilus*, *Bacillus subtilis*, and *Saccharomyces cerevisiae* were used for fermentation. The concentration of strains used in this study was *Lactobacillus acidophilus*, 1.5 × 10^9^ CFU (counting flora unite)/mL; *Bacillus subtilis*, 5.6 × 10^8^ CFU/mL; *Saccharomyces cerevisiae*, 2 × 10^8^ CFU/mL. The ratio of the mixed strains for FRSM was *Lactobacillus acidophilus*: *Bacillus subtilis*: *Saccharomyces cerevisiae* in the ratios: 1:3:2. The fermentation conditions were constant temperature 33 °C, feed water ratio 1:1, time 84 h, and inoculum volume 6%. Ultimately, FRSM was dried for 3 days at 55 °C, and then the dried samples were ground to pass through a 0.5 mm sieve, and kept at room temperature until mixed in the diets. The total dry matter losses were about 8% in our previous study. All of the processes were completed in the Nankou Experiment Base of the Chinese Academy of Agricultural Sciences. The mixing machine, a constant temperature fermented room, and the fermentation bag (25 kg/bag), with one way degassing valve, were used during the fermentation process. The total cost was in the range of $22–30 USD/ton. 

### 2.3. Experiment 1

A total of 72 Arbor Acres (AA) male broilers (45 ± 3 g/bird) were used to determine the AME and AMEn values of RSM and FRSM. All of the birds were raised together from 0 to 21 day and received a starter maize–soybean meal basal diet. The basal diet was formulated to meet or exceed the National Research Council (NRC) (1994) [17] energy and nutrient requirements of poultry. On day 21, the birds were allocated to 3 treatments in a completely randomized design. Each treatment had 6 replicate cages with 4 birds per cage. The mean body weight among replicates was 2904 ± 35 g. The AME and AMEn values of the RSM and FRSM were determined by the substitution method. A common maize–soybean meal diet served as the reference diet to meet or exceed NRC (1994) energy and nutrient requirements of poultry. The RSM and FRSM samples proportionally replaced 30% of the energy-yielding components of the basal diet. Titanium dioxide (TiO_2_) was added to diets at 0.4%, as an exogenous digestibility marker. The proportional replacement of the energy-yielding components was essential for the calculation of AME for FRSM and RSM [18,19]. Table 2 shows the ingredients and chemical compositions of the reference and experimental diets. All birds were reared under identical conditions according to the Arbor Acres broiler management guide [20]. Birds were raised on floor monolayer cage (1.2 × 0.9 × 0.7 m) and they had free access to clean water and feed for the entire experiment period. During the first 3 day, the ambient temperature in the room was maintained at 35 °C and was gradually reduced, reaching 25 °C at 21 day of age. The lighting program was a period of 17 h of light and 7 h of darkness.

All of the broilers were fed the same basal diet for 21 day and were subsequently provided with the experimental diets. After a diet acclimation period from 22 to 25 day, total excreta was collected daily from 26 to 30 day and stored in a −20 °C freezer. Spilled feed and feathers were removed from the excreta samples. At the end of excreta collection, all of the excreta were dried, ground, passed through a 0.5 mm screen, and stored in a 4 °C freezer prior to analysis. The average daily feed intake for each treatment was recorded.

### 2.4. Experiment 2

A total of 72 AA male broilers were used to determine the apparent ileal digestibility (AID) and SID of the amino acids in RSM and FRSM. All of the birds were raised together from 0 to 25 day and received maize–soybean basal diet. The basal diet was formulated to meet or exceed the NRC (1994) energy and nutrient requirements of broilers. On day 25, the birds were allocated to 3 treatments in a completely randomized design. Each treatment had 6 replicate cages with 4 birds per cage, the mean body weight among replicates was 3509 ± 61 g. A nitrogen-free diet (NFD) was formulated to determine the endogenous amino acids losses (EAAL) [21]. The proportion of RSM and FRSM in the diets was adjusted with maize starch (CP < 3%), microcrystalline cellulose, saccharose, soybean oil, limestone, salt, and vitamin–mineral premix to maintain a 20% CP in all diets. TiO2 was added to the diets, at a rate of 0.4%, as an exogenous digestibility maker. The amino acids composition of RSM and FRSM is presented in Table 3. The ingredients and chemical composition of diets are shown in Table 4. The bird husbandry conditions were the same as in Experiment 1.

On day 25, the birds were acclimatized to their new environments and diets during the first 48 h. After 48 h of ad libitum feed intake, the birds were deprived of food for 6 h and then resumed feed intake ad libitum. Three hours later, all birds were euthanized by intravenous injection of pentobarbitone. The contents of the ileum from the Meckel’s diverticulum to a point approximately 50 mm anterior to the ileocecal junction were collected in a plastic culture dish. Ileal digesta of all birds collected in a cage were pooled in the same plastic culture dish, immediately stored at −20 °C, and subsequently freeze-dried. Dried ileal digesta were then ground using a micro grinding machine to pass through a 0.5-mm sieve and stored in plastic tubes at −4 °C for subsequent chemical analyses.

### 2.5. Chemical Analysis

Samples of RSM, FRSM, diets, excreta, and ileal digesta were dried at 105 °C in a drying oven to determine the dry matter (DM) content (method 4.1.06; AOAC (Association of Official Analytical Chemists) 2000) [22]. RSM and FRSM were analyzed for glucosinolates [23], crude fiber (Method 978.10, AOAC 2006) [24], crude fat (method 920.39; AOAC 2006) [24], polypeptides (GB/T 22492-2008) [25], and L-lactic acid [26]. Total nitrogen content was determined with a combustion analyzer (Dumatherm, Gerhardt, Germany). Crude protein was calculated as N × 6.25. The gross energy contents of diets and excreta were determined in a bomb calorimeter (C2000, IKA, Guangzhou, China) using benzoic acid as the calibration standard. The amino acids of RSM, FRSM, diets, and ileal digesta were determined by an automatic amino acid analyzer (Hitachi L-8800, Tokyo, Japan). Samples for non-sulfur amino acid analysis were hydrolyzed using pretreatment with 6 M HCl for 24 h. The pH of the hydrolysate was adjusted to 2.20, centrifuged, and filtered [27]. Methionine and cystine were determined as methionine sulfone and cysteic acid after cold performic acid oxidation overnight and hydrolyzing with 7.5 N HCl at 110 °C (procedure 4.1.11; alternative 1; AOAC, 2000) [22] for 24 h, followed by analysis using an amino acid analyzer (Hitachi L-8800, Tokyo, Japan). The TiO2 content of diets and ileal digesta was analyzed according to the method described by Titgemeyer et al. (2001) [28]. All chemical analyses were conducted with three repeats, and the average values were used in the statistical analysis.

### 2.6. Calculations

The AME and AMEn values of RSM and FRSM were determined by the substitution method using the following equations [19,29]:ADMD = (Mconcdg − Mconfd)/Mconcdg,(1)
where ADMD means the apparent digestibility of DM in diets. Mconcdg and Mconfd are feces marker and diet marker (TiO_2_), g/kg, DM;
ME (MJ/kg) = Econfd − (1 − ADMD) × Econcdg,(2)
MEn (MJ/kg) = AMED − RN × 8.22.(3)

ME is the apparent metabolizable energy of a diet. MEn is the apparent metabolizable energy of diets corrected by N. Econfd and Econcdg are the energy of feed and excreta, MJ/kg, DM. RN is total nitrogen retained to correct the AME. The correction factor for broilers was 8.22. The RN was calculated as:RN = Nfd − (Tofd/Tofc) × Nfc(4)
where Nfd is nitrogen/g feed; Tofd is TiO_2_/g feed; Tofc is Tio2/g feces; Nfc is nitrogen/g feces. 

The AME and AMEn of the RSM and FRSM samples were calculated as
AME (MJ/kg) = MEb + (Met − MEb)/f,(5)
where AME is the apparent metabolizable energy of RSM and FRSM (MJ/kg, DM), MEb is the apparent metabolizable energy of basal diet, MEt is the apparent metabolizable energy of test diet, and f is the percentage of substitution rate. The AMEn of RSM and FRSM samples were calculated using the same equation.

The AID and SID of RSM and FRSM were determined by the NFD method using the following equations described by Ullah et al. (2017) [21].
Apparent ileal amino acid digestibility (AID), % = (1 − (Marker in diet/Marker in ileal digesta) × (Amino acid in ileal digesta/Amino acid in diet))(6)

The endogenous ileal amino acid losses were used to calculate the SID of amino acids by the following equations; the marker in diet and ileal digesta wasTiO_2_;
Ileal amino acid flow (mg/kg, DM intake) = ((Amino acid in ileal digesta) × (Diet marker/Ileal marker)),(7)
Standardized ileal amino acid digestibility (SID), % = AID + (Ileal amino acid flow/Amino acid in raw material).(8)

### 2.7. Statistical Analysis

The data for both experiments were analyzed using SPSS 19.0 statistical software (2010, SPSS Inc., Chicago, IL, USA) under a completely randomized design. Significant differences among treatments were determined using an independent sample t-test. Data are presented as means ± SEM. Statistical significance was set at *p* < 0.05.

## 3. Results

The chemical compositions of RSM and FRSM are presented in Table 1. The fermentation of RSM by microorganisms increased the content of CP (370.5 vs. 409.0 g/kg). The amino acid composition of RSM and FRSM shows that fermentation increased the content of several amino acids in rapeseed meal, especially Asp, Thr, Ser, Glu, Pro, Ala, and Lys (Table 3). The greatest change caused by fermentation was for glucosinolates, which decreased from 36.08 to 17.09 µmol/g on a DM basis. The fermentation of RSM increased the content of polypeptides (from 8.4 to 21.5 g/kg) and increased the content of L-lactic acid (from 10.1 to 55.8 g/kg) compared to unfermented RSM. The concentration of crude fat was similar in RSM and FRSM (43.1 and 43.9 g/kg) (Table 1).

The data for AME and AMEn of RSM and FRSM are shown in Table 5. Moreover, the AME and AMEn of the basal diet can be seen in Table 6. FRSM had higher (*p* < 0.05) AME and AMEn values than RSM. The fermentation of RSM increased AME from 7.44 to 8.51 MJ/kg (*p* = 0.04) and AMEn from 7.17 to 8.26 MJ/kg (*p* = 0.03).

In this experiment, the nitrogen-free diet (NFD) was formulated to determine endogenous amino acids losses (EAAL). Table 7 is the concentration of endogenous amino acids losses. Results of AID and SID of amino acids in FRSM and RSM fed to broilers are shown in Table 8. The AID and SID of various amino acids in rapeseed meal were increased after fermentation. A significant difference between FRSM and RSM was observed on AID for Ala, Val, Ile, Leu, Tyr, Lys, Arg, and Phe. In addition, feeding on FRSM resulted in higher (*p* < 0.05) SID of Ala, Val, Ile, Leu, Tyr, Lys, Arg, and Phe. No significant differences were observed between RSM and FRSM on the AID of Asp, His, Thr, Ser, Glu, Pro, Gly, Met, and Cys, and the SID of Asp, His, Thr, Ser, Glu, Pro, Gly, Met, and Cys.

## 4. Discussion

In modern commercial poultry farming systems, the production of broiler feed contributes up to 70% of the total production cost. Due to increases in the global feed prices, there is now a tendency in the poultry industry to develop new, high-quality protein sources, which can maximize animal growth performance and maintain body health [13,30]. FRSM is an ideal feed ingredients substitute for soybean in commercial poultry production. Owing to its low price, there is increasing interest in incorporating FRSM into broiler rations to take advantage of its positive influences, particularly in production parameters and gut health [31,32]. In the present study, strains of *Lactobacillus acidophilus, Bacillus subtilis,* and *Saccharomyces cerevisiae* were used for the fermentation of RSM. Previous studies have demonstrated that fermentation can improve the quality of RSM by reducing the levels of antinutritional factors [33]. Vig and Walia [34] showed that a 10 d fermentation reduced the glucosinolates content by 43.1% in RSM. We also found that fermentation changed the chemical composition of RSM. The reduction of glucosinolates and other anti-nutritional factors during fermentation may be due to their degradation by microbial enzymes [35]. In the current study, we noted a minor improvement in the CP content of FRSM compared with RSM, which is consistent with the findings of Drazbo et al. [4]. According to Hu et al. [23], the increase in CP content is mostly associated with a decrease in the concentrations of non-structural carbohydrates. This reflects changes in dry matter content rather than an actual increase in protein content. Moreover, we found that fermentation increased the gross energy in FRSM. Drazbo et al. [4] also noted that the gross energy of RSM was increased from 21.45 to 21.92 MJ/kg after fermentation. The increase in gross energy in FRSM may reflect an increase in the levels of crude protein. The gross energy of crude protein is higher than that of non-structural carbohydrates, which may explain the higher gross energy in FRSM.

Previous studies also reported that fermentation can increase the level of polypeptides in RSM, which is consistent with our finding [36,37]. The protein structure of RSM was changed during the fermentation process, especially in the contents of polypeptides. As reported by Windey et al. [38], the degradation of proteins starts with hydrolysis of the proteins to smaller peptides by bacterial proteases and peptidases; thus, the microbial enzymes produced during the fermentation process may have played a major role in the increase of polypeptides in FRSM. Jakobsen et al. [39] found that fermented RSM had no effect on the content of crude fat and ether extracts, which is in agreement with our results. Insoluble fiber and fat fractions are more resistant to fermentation than their soluble counterparts; protein and digestible starch are preferred as nutrient sources during the fermentation process [29].

The AME values of RSM and FRSM were consistent with data from the Feed Composition and Nutritive Values in China [40] (RSM, AME, 7.41 MJ/kg). FRSM achieved higher AME and AMEn values. The precise reasons for fermentation improves the RSM AME and AMEn values in broilers are not yet clear; however, some hypotheses have been advanced. The availability of energy in RSM and FRSM mainly depends on the balance of the energy-yielding constituents in the feedstuff and factors that impede their utilization. The contents of crude fiber, acid detergent fiber, and neutral detergent fiber were reduced a lot in the present experiment if we take dry matter losses into consideration (about 8%). Consequent reduction in crude fiber content has been reported to increase ME of RSM [41]. The amounts of ingredients, such as pectic oligosaccharides and insoluble fibers in RSM, may also reduce the energy digestibility [42]. In this study, during *Lactobacillus acidophilus* fermentation of rapeseed meal, a considerable amount of amylases, proteases, phytases, and *β*-glucanases may be produced and activated. These enzymes consumed the insoluble fibers in RSM and produced soluble matters with simple structure; carbohydrate composition was modified as a result of microbial metabolism and, to some extent, due to activation of inherent enzymes in the cereal protein feedstuff [43]. In a study by Alahyarishahrasb et al. [29], enzyme treatment (*β*-glucanase, 0.5 g/kg) barley had significantly higher AMEn values than no enzyme treatment when fed to cockerels. Studies have shown that the addition of enzymes to chicken diets degrades endosperm cell walls, resulting in increased digestibility of nutrients, which otherwise may be encapsulated in the cell structures [44,45]. In addition, a chemical analysis showed that fermentation reduced the contents of antinutrients in FRSM, which may directly improve the digestibility of nutrients. There is no direct evidence showing the relationship between glucosinolates and AME values, but it has been reported that reduced glucosinolates in FRSM can improve broiler health status, which may have a positive effect on energy utilization in broiler chickens [4,12,21]. Phytate is another antinutrient in animal feedstuffs, but failed to measure in this study. Selle et al. [46] observed that through degradation of phytate, Phy prevented phytate-protein complexes, with a subsequent increase in nutrient digestibility. Some studies also suggest that FRSM increases the intestinal length index and maintains a normal gut microbial ecosystem [23]. Healthy intestinal ecology is essential for broilers to digest and absorb nutrients. The high concentrations of L-lactic acid during the fermentation process has beneficial effects on intestinal morphology and controlling pathogens in animal digestion systems, and, thus, finally improves the utilization of energy [32,47]. The fermentation process also changes the structure of large protein molecules, increases the content of polypeptide in FRSM, and improves its protein solubility and biological utilization ability. Fermentation microbes can use the inferior proteins in RSM to synthesize microbial cell proteins. According to our knowledge, microbial cell protein can be efficiently utilized by broiler chickens, and the biological value of microbial cell protein is similar to soybean meal, but is higher than RSM [48]. Therefore, it is the combination of changed antinutrients, crude protein, gross energy, and L-lactic acid that ultimately influenced the AME and AMEn values of FRSM.

The information (about the digestibility of amino acids in FRSM) is limited, and the values for the AID and the SID of amino acids in FRSM is also not known. In the present study, we used a nitrogen-free diet method to correct the endogenous amino acid flux in broilers. Our results indicated that broiler digestibility of amino acids was improved when they were fed FRSM diets in comparison with RSM. The increased AID and SID of amino acids in FRSM, compared with RSM, are likely due to the reduced anti-nutritional factors, low-molecular-size proteins, and L-lactic acid contents in FRSM. Ullah et al. [21] showed that RSM with reduced glucosinolates and erucic acid (00-RSM) showed a greater SID for all amino acids, except Arg, His, Phe, Cys, and Glu, compared with RSM. Moreover, many studies demonstrated that the presence of phytate negatively interfered with gastric digestion and the concomitant intestinal tract response [49,50]. Selle et al. [51] observed a greater phytate content and lower AID of Cys, Thr, Pro, and Gly of broilers. In a report by Cowieson and Ravindran [49], the increased protein digestibility by phytase supplementation was associated with the reduction of endogenous protein flow and nutrition losses. Gallardo et al. [43] also noticed the reduced phytate decreased the endogenous inputs for protein digestion, which resulted in an improvement in SID of amino acids, principally Thr, Asp, and Gly, which predominate in the endogenous proteins [52]. This may be the same way FRSM achieved higher AID and SID of Ala, Val, Ile, Leu, Tyr, Lys, Arg, and Phe observed in the present study. Moreover, the increased polypeptide of FRSM by fermentation, when compared to RSM, suggest an increase in the degree of protein hydrolysis; hence, increasing the proportion of soluble low-molecular-size proteins, thereby, making the protein more available for uptake by the chickens [53]. These observations suggest that the fermentation process increased the levels of functional factors (polypeptides, L-lactic acid, and other organic acids) in FRSM, which may be responsible for the improvement of digestion and absorption of proteins and higher amino acid digestibility. Therefore, further research will focus on the specific components of these functional factors and their effects.

## 5. Conclusions

Fermentation had a significant effect on the chemical composition of RSM. It reduced the concentration of glucosinolates and increased the concentrations of CP, gross energy, polypeptides, and lactic acid. In comparison to RSM, FRSM had greater AMEn values and SID of amino acids. Therefore, FRSM was nutritionally superior to RSM for use in broiler diets. The basic data and theories for the research of fermented rapeseed meal in this study would provide a reference for the application of fermented feed in poultry nutrition.

## Figures and Tables

**Table 1 animals-10-01774-t001:** Chemical composition of fermented rapeseed meal (FRSM) and rapeseed meal (RSM) used in this study (%, dry matter basis).

Items	Crude Protein	Glucosinolates µmol/g	Polypeptides	Lactic Acid	Crude Fiber	Acid Detergent Fiber	Neutral Detergent Fiber	Crude Fat	Gross Energy (KJ/kg)
RSM	37.05 ± 0.56	36.08 ± 0.68	0.84 ± 0.09	1.01 ± 0.13	17.47 ± 0.54	26.32 ± 0.16	3 3.89 ± 0.34	4.31 ± 0.19	20.60 ± 0.15
FRSM	40.90 ± 0.43	17.09 ± 0.32	2.15 ± 0.12	5.58 ± 0.52	16.72 ± 1.72	24.15 ± 0.22	31.27 ± 0.43	4.39 ± 0.05	21.19 ± 0.13

Note: Determined in triplicate.

**Table 2 animals-10-01774-t002:** Dietary composition and calculated analyzed nutrient levels of the test diets used for the apparent metabolizable energy (AME) and greater nitrogen-corrected apparent metabolizable energy (AMEn) experiment.

Ingredients %	Basal Diet	RSM Diet	FRSM Diet
Maize	56.18	38.57	38.57
Soybean meal	30.57	20.99	20.99
RSM	0.00	30.00	0.00
FRSM	0.00	0.00	30.00
Maize gluten meal	3.97	2.72	2.72
Soybean oil	4.98	3.42	3.42
CaHPO_4_	1.73	1.73	1.73
Met	0.12	0.12	0.12
*L*-Lys-His	0.06	0.06	0.06
Limestone	1.21	1.21	1.21
NaCl	0.27	0.27	0.27
Choline chloride	0.01	0.01	0.01
TiO_2_	0.40	0.40	0.40
Vitamin Premix ^1^	0.02	0.02	0.02
Minerals Premix	0.20	0.20	0.20
Zeolite Powder	0.28	0.28	0.28
Total	100.00	100.00	100.00
Nutrient levels ^2^			
ME (MJ/kg)	12.96	11.35	11.35
CP	20.26	25.12	25.84
Ca	0.90	1.04	1.04
AP	0.41	0.46	0.46
Met	0.42	0.47	0.48
Lys	0.97	1.06	1.08

Note: Abbreviation(s): RSM, rapeseed meal; FRSM, fermented rapeseed meal; ME, metabolizable energy; CP, crude protein; AP, available phosphorus; Met, Methionine; Lys, Lysine; FRSM, rapeseed meal; RSM, rapeseed meal. ^1^ The premix provided the following (per kg) of diets: vitamin A 10,000 IU, vitamin D_3_ 2000 IU, vitamin E 20 IU, vitamin B_1_ 2.0 mg, vitamin K_3_ 2.5 mg, vitamin B_2_ 4.0 mg, vitamin B_6_ 5.0 mg, vitamin B_12_ 0.02 mg, *D*-pantothenic acid 11.0 mg, nicotinic acid 35 mg, folic acid 0.5 mg, biotin 0.12 mg, Fe (as ferrous sulfate) 80 mg, Cu (as copper sulfate) 8 mg, Zn (as zinc sulfate) 78 mg, Mn (as manganese sulfate) 100 mg, I (as potassium iodide) 0.34 mg, Se (as sodium selenite) 0.15 mg. ^2^ Crude protein, Met, Lys are analyzed values, ME, Ca and AP are calculated values.

**Table 3 animals-10-01774-t003:** Amino acids composition of fermented rapeseed meal (FRSM) and rapeseed meal (RSM) (dry matter basis).

Amino acids %	RSM	FRSM
Crude protein	37.05	40.90
Indispensable amino acids		
Arg	2.21	2.29
Gly	1.98	2.14
His	1.07	1.14
Ile	1.63	1.73
Leu	2.81	3
Lys	2.43	2.63
Met	0.84	0.88
Phe	1.51	1.62
Pro	2.42	2.93
Thr	1.65	1.79
Val	2	2.23
Dispensable amino acids		
Ala	1.73	1.99
Asp	2.69	2.83
Cys	0.96	1.02
Glu	6.35	6.99
Ser	1.48	1.64
Tyr	1.05	1.1

Note: Abbreviation(s): Arg, Arginine; Gly, Glycine; His, Histidine; Ile, Isoleucine; Leu, Leucine; Lys, Lysine; Met, Methionine; Phe, Phenylalanine; Pro, Proline; Thr, Threonine; Val, Valine; Ala, Alanine; Asp, Asparagine; Cys, Cysteine; Glu, Glutamic acid; Ser, Serine; Tyr, Tyrosine. The same as blows. Determined in triplicate.

**Table 4 animals-10-01774-t004:** Dietary composition and nutrient levels of the experimental diets for the apparent ileal digestibility (AID) and standardized ileal digestibility (SID) experiment.

Ingredients %	NFD	RSM Diet	FRSM Diet
RSM	0	53.82	0
FRSM	0	0	48.62
Maize starch	68.10	28.0	33.20
Saccharose	19.98	10.00	10.00
Crystalline cellulose	5.00	0	0
Soybean oil	3.0	3.50	3.50
CaHPO_4_	1.9	1.95	1.95
Limestone	1.0	1.33	1.33
NaCl	0.3	0.30	0.30
Choline chloride (mg/kg)	0.1	0.20	0.20
TiO_2_	0.4	0.40	0.40
Vitamin Premix ^1^	0.02	0.02	0.02
Minerals Premix	0.20	0.20	0.20
Zeolite Powder	0	0.28	0.28
Total	100.00	100.00	100.00
Nutrient levels ^2^			
ME (MJ/kg)	13.31	10.22	9.58
CP	0.13	19.88	20.09
Ca	0.80	0.90	0.92
AP	0.35	0.41	0.42

Note: Abbreviation(s): NFD, nitrogen-free diet; RSM, rapeseed meal; FRSM, fermented rapeseed meal; ME, metabolizable energy; CP, crude protein; AP, available phosphorus. ^1^ The premix provided the following per kg of diets: Vitamin A 10,000 IU, Vitamin D3 2000 IU, Vitamin E 20 IU, Vitamin B1 2.0 mg, Vitamin K3 2.5 mg, Vitamin B2 4.0 mg, Vitamin B6 5.0 mg, Vitamin B12 0.02 mg, D-pantothenic acid 11.0 mg, nicotinic acid 35 mg, folic acid 0.5 mg, biotin 0.12 mg, Fe (as ferrous sulfate) 80 mg, Cu (as copper sulfate) 8 mg, Zn (as zinc sulfate) 78 mg, Mn (as manganese sulfate) 100 mg, I (as potassium iodide) 0.34 mg, Se (as sodium selenite) 0.15 mg. ^2^ Crude protein, Met, Lys are analyzed values, ME, Ca and AP are calculated values.

**Table 5 animals-10-01774-t005:** Metabolizable and nitrogen corrected metabolizable energy of RSM and FRSM (dry matter basis).

Items	RSM	FRSM	SEM	*p*-Values
AME, MJ/kg	7.44 ± 0.29	8.51 ± 0.11	0.26	0.04
AMEn, MJ/kg	7.17 ± 0.24	8.26 ± 0.11	0.21	0.03

Note: abbreviation(s): RSM, rapeseed meal; FRSM, fermented rapeseed meal; SEM, standard error of the mean; AME, apparent metabolizable energy values; AMEn, nitrogen-corrected apparent metabolizable energy.

**Table 6 animals-10-01774-t006:** Metabolizable and nitrogen corrected metabolizable energy of basal diet (dry matter basis).

Items	Basal Diet	RSM Diet	FRSM Diet	SEM	*p*-Values
AME, MJ/kg	13.45 ± 0.31	11.59 ± 0.16	11.90 ± 0.09	0.31	0.05
AMEn, MJ/kg	12.69 ± 0.28	10.97 ± 0.13	11.29 ± 0.10	0.26	0.04

Note: abbreviation(s): RSM, rapeseed meal; FRSM, fermented rapeseed meal; SEM, standard error of the mean; AME, apparent metabolizable energy values; AMEn, nitrogen-corrected apparent metabolizable energy.

**Table 7 animals-10-01774-t007:** Concentration of endogenous amino acid losses used to standardize the amino acid digestibility.

Amino Acids	Endogenous Amino Acid Concentration (mg/kg Dry Matter Intake)
Indispensable amino acids	
Arg	291.10 ± 7.63
Gly	184.33 ± 8.52
His	90.75 ± 3.29
Ile	122.51 ± 6.52
Leu	200.74 ± 11.23
Lys	213.10 ± 7.58
Met	85.62 ± 2.47
Phe	255.74 ± 10.37
Pro	375.75 ± 18.41
Thr	445.02 ± 13.96
Val	201.62 ± 8.56
Dispensable amino acids	
Cys	107.64 ± 5.28
Ala	155.59 ± 7.41
Asp	554.54 ± 19.56
Glu	442.66 ± 17.63
Ser	418.14 ± 19.36
Tyr	192.15 ± 8.69

**Table 8 animals-10-01774-t008:** Measure values of AID and SID for amino acids in RSM and FRSM (%, dry matter basis).

Amino Acids	AID %			SID %		
	CRSM	FRSM	*p*-Values	CRSM	FRSM	*p*-Values
Indispensable amino acids						
Arg	76.41 ± 1.21	80.13 ± 1.03	0.05	78.85 ± 1.22	82.80 ± 1.04	0.03
Gly	63.28 ± 1.42	66.48 ± 1.14	0.12	65.03 ± 1.42	68.22 ± 1.14	0.12
Pro	70.66 ± 1.15	68.82 ± 1.13	0.31	73.15 ± 1.15	71.60 ± 1.13	0.38
Thr	56.70 ± 1.64	60.73 ± 1.26	0.09	61.64 ± 1.62	65.78 ± 1.27	0.08
His	74.06 ± 1.34	73.32 ± 0.83	0.67	75.58 ± 1.34	75.04 ± 0.83	0.75
Ile	64.65 ± 1.45	72.27 ± 1.21	0.01	66.11 ± 1.46	73.71 ± 1.21	0.01
Leu	66.79 ± 1.27	73.12 ± 1.34	0.01	68.13 ± 1.28	74.47 ± 1.34	0.01
Lys	67.29 ± 0.98	73.13 ± 1.09	<0.01	69.00 ± 0.98	74.83 ± 1.09	<0.01
Met	79.87 ± 1.01	82.32 ± 1.25	0.15	81.74 ± 1.01	84.35 ± 1.25	0.14
Phe	69.22 ± 1.35	74.51 ± 1.56	0.04	72.37 ± 1.36	77.68 ± 1.57	0.04
Val	63.64 ± 0.10	70.38 ± 1.28	0.01	65.50 ± 1.10	72.29 ± 1.28	0.01
Dispensable amino acids						
Cys	61.31 ± 1.61	60.95 ± 0.08	0.85	63.42 ± 1.62	63.29 ± 0.83	0.94
Ala	65.96 ± 1.13	73.52 ± 1.13	<0.01	67.59 ± 1.13	75.10 ± 1.13	<0.01
Asp	58.24 ± 1.75	63.09 ± 1.50	0.06	62.06 ± 1.75	67.07 ± 1.50	0.06
Glu	77.04 ± 1.21	78.62 ± 1.83	0.33	78.31 ± 1.21	79.95 ± 0.83	0.31
Ser	60.96 ± 1.71	62.52 ± 1.22	0.48	65.97 ± 1.71	67.63 ± 1.22	0.45
Tyr	64.30 ± 1.54	68.02 ± 1.79	0.01	67.62 ± 1.54	71.49 ± 1.79	0.02

Note: abbreviation(s): AID, apparent ileal digestibility; SID, standardized ileal digestibility; RSM, rapeseed meal; FRSM, fermented rapeseed meal; DM, dry matter; SEM, standard error of the mean.

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
