# Peer review of "Effects of Fermentation on Standardized Ileal Digestibility of Amino Acids and Apparent Metabolizable Energy in Rapeseed Meal Fed to Broiler Chickens"

_animals, 2020, doi:10.3390/ani10101774_

Round 1

Reviewer 1 Report

The topic of the manuscript is interesting and fits well within the scope of the journal. There have been several previous publications on this topic but this one still adds to the existing knowledge.

While the methods used seems to be appropriate and correctly done I have severe problem with this manuscript as it is a real sloppy prepared manuscript and contains several problems which restrain a publication in its present form

In addition , the discussion is very bad and gives me the feeling that the authors have real problems explaining their results especially the mode of action why ME and SID is improved.

For specific comments see the marked section in the attached manuscript!!

Author Response

Thanks very much for your kind work and consideration on publication of our manuscript entitled “Effects of fermentation on standardized ileal digestibility of amino acids and apparent metabolizable energy in rapeseed meal fed to broiler chickens” (Manuscript ID animals-876031). Those comments are all valuable and very helpful for revising and improving our paper, as well as the important guiding significance to our researches. We have revised the manuscript according to your comments and suggestions, and the amendments are marked with red in the revised manuscript. Below you will find our point-by-point responses to your comments and questions. The whole manuscript has been carefully checked again by professional English-language editor and ourselves. 

We do hope we could understand your questions correctly and have given right answers in the revised manuscript. Please feel free to inform me if there are still any questions. Thank you very much!

Reviewer 2 Report

Dear Authors,

Manuscript (animals-876031) describes interesting issues regarding the effect of RSM fermentation on AME and SID of AA in broilers fed fermented RSM. Two experiments were performed on male Arbor Acres chickens. The Authors suggest on the basis of the results that fermentation of RSM can improve AME and digestibility of amino acids in RSM. I read with interest this manuscript. This is a really interesting paper and results show new data.

However, the manuscript needs some corrections.

In general:

Please discuss in text the feasibility under field conditions and possible costs.

Why you used NRC, 1994 ? Instead it was better to use the breed catalogue which is more recent - the latest Arbor Acres Broiler Nutrition Specifications by Aviagen (2019) or the previous from 2014 (if the experiment was run in the previous year).

Why did you decide only to use one gender in your study?

Why in this study results of ADFI, ADG or FCR are not showed? It would be interesting if the birds reached/exceeded the weight/FCR/daily intake yield prescribed by the breeding company (Aviagen). Please provide information and discuss.

Please explain in text why only male chicks were housed and evaluated?

Why did the authors rear the broilers in cages rather than pens? It is always preferable in terms of animal welfare. Same consideration for the lighting programme (23L:1D): according to the welfare regulations, a minimum of 6 hours of darkness is recommended. How does applied lighting programme comply with the breeding company recommendations? Please, explain.

Detailed comments:

P2 paragraph 2 Line 168: Does the animal ethics review committee provide a reference/case number you can share in the text?

P2 paragraph 2.2 Lactobacillus acidophilus, Bacillus subtilis, and Saccharomyces cerevisiae – italic

P3L1 – Arbor Acres

Table 2 correct CRSM to RSM

Table 3 superscript 1 missing in the table

P3L6 Please provide information about pen size and stocking density in kg/m2.

P7L10 – who checked physiological state of the test birds? Veterinarian ?

P8L12 correct “dray” to “dry”

P9L3 reference needed

P9L23 remove” Hu et al., 2016; Zhang et al., 2016”

P9L27-29 (“The biological value of microbial cell….”)  reference needed

P10 References – minor typo errors in references – please read carefully

Ref 3 - Joanna, Mirosław, Sebastain, Andrzej, Anna and Katarzyna are the first names, not surnames; please see reference and correct.

Ref 4 – this work was published in Poult. Sci., please correct

Each treatment had 6 replicate cages. Why analyses were performed in triplicate ?

Author Response

(The authors gave the same response as above.)

Reviewer 3 Report

Effects of fermentation on standardized ileal digestibility of amino acids and apparent metabolizable energy in rapeseed meal fed to broiler chickens

Major concerns

  1. Grammars and writings should be corrected and revised thoroughly. Here are some examples in the Abstract section.

Ex.in simple summary, line 5-6,” has been shown as an effective way…”, line 7 “this improvement is not …”, line 8, “.. value of FRSM is not very well-studied”.

  1. in Abstract, line 8-9 , “Results show that fermentation …”., in line 16 “and therefore was nutritionally superior…”
  2. Introduction line 5-7, “ Glucosinolates and its secondary metabolites are considered as toxic agents which affect growth performance and health status of animals, and thus is recommend no more than 10% of rapeseed meal ( Glucosinolates: < 25 μmol/g)in broilers diet.
  3. The authors should enclose the results of growth performance such as BW gain and FCR to confirm the superiority of FRSM.
  4. Lactobacillus acidophilus, Bacillus subtilis, and Saccharomyces cerevisiae were used for fermentation. So, how the authors excluded the effects from bacteria as probiotics or prebiotics instead of the effects of RSM components?

Author Response

Thanks very much for your kind work and consideration on publication of our manuscript entitled “Effects of fermentation on standardized ileal digestibility of amino acids and apparent metabolizable energy in rapeseed meal fed to broiler chickens” (Manuscript ID animals-876031). Those comments are all valuable and very helpful for revising and improving our paper, as well as the important guiding significance to our researches. We have revised the manuscript according to your comments and suggestions, and the amendments are marked with red in the revised manuscript. Below you will find our point-by-point responses to your comments and questions. The whole manuscript has been carefully checked again by co-author  Shoaib Ahmed Pirzado.

We do hope we could understand your questions correctly and have given right answers in the revised manuscript. Please feel free to inform me if there are still any questions. Thank you very much!

Reviewer #1:

Question 1. Major concerns

  1. Grammars and writings should be corrected and revised thoroughly. Here are some examples in the Abstract section.

Ex.in simple summary, line 5-6,” has been shown as an effective way…”, line 7 “this improvement is not …”, line 8, “.. value of FRSM is not very well-studied”.

  1. in Abstract, line 8-9 , “Results show that fermentation …”., in line 16 “and therefore was nutritionally superior…”
  2. Introduction line 5-7, “ Glucosinolates and its secondary metabolites are considered as toxic agents which affect growth performance and health status of animals, and thus is recommend no more than 10% of rapeseed meal ( Glucosinolates: < 25 μmol/g)in broilers diet.

Answer: Thank you very much for your precious encouragement and suggestions. We totally agree with you and we re-prepared this manuscript carefully. We are so sorry to describe the discussion part unclear. We have revised the manuscript according to your comments and suggestions, and the amendments are singed in the revised manuscript.

  1. The authors should enclose the results of growth performance such as BW gain and FCR to confirm the superiority of FRSM.

Answer: Thanks very much for your question. This experiment was designed to use a TiO2 marker method, the experiment diets were specially formulated, especially the NFD ( nitrogen-free diet) was nearly no protein. Besides, the official test just lasted few days, so, I think the feed intake and body weight gain didn’t means so much for the experiment. Some published paper also contains no data about feed intake or body weight gain(Nitrogen-corrected apparent metabolizable energy values of barley varies by treatment and species). we designed another experiment specially to explore the effect of fermented rapeseed meal on the growth performance and immune status in broilers.Hope you understand why I didn’t provide feed intake data.

  1. Lactobacillus acidophilus, Bacillus subtilis, and Saccharomyces cerevisiae were used for fermentation. So, how the authors excluded the effects from bacteria as probiotics or prebiotics instead of the effects of RSM components?

Answer:  In this experiment, FRSM was dried for 3 days at 55°C, and then the dried samples were ground to pass through a 0.5 mm sieve. Most of probiotics lose their activity under this adverse environment. Besides, the key to the probiotic effect of probiotics is to colonize the animal intestines, the official test in this study just lasted few days, so we don’t think the mainly effect comes from bacteria.

Reviewer 4 Report

The results of the study suggested fermentation a way to improve the nutritional value (AME and AA dig.) of rapeseed meal. Would be constructive to add a few lines in discussion section to share Authors' view on feasibility of this technique on a large-, commercial-scale.  

Author Response

Thanks very much for your kind work and consideration on publication of our manuscript entitled “Effects of fermentation on standardized ileal digestibility of amino acids and apparent metabolizable energy in rapeseed meal fed to broiler chickens” (Manuscript ID animals-876031). Those comments are all valuable and very helpful for revising and improving our paper, as well as the important guiding significance to our researches. We have revised the manuscript according to your comments and suggestions, and the amendments are marked with red in the revised manuscript. Below you will find our point-by-point responses to your comments and questions. The whole manuscript has been carefully checked again by co-author  Shoaib Ahmed Pirzado.

We do hope we could understand your questions correctly and have given right answers in the revised manuscript. Please feel free to inform me if there are still any questions. Thank you very much!

Reviewer #1:

Question 1. The results of the study suggested fermentation a way to improve the nutritional value (AME and AA dig.) of rapeseed meal. Would be constructive to add a few lines in discussion section to share Authors' view on feasibility of this technique on a large-, commercial-scale.  

Answer: Thank you very much for your precious encouragement and suggestions. We totally agree with you and we re-prepared this manuscript carefully. We are so sorry to describe the discussion part unclear. We have revised the manuscript according to your comments and suggestions, and the amendments are singed in the revised manuscript.

We added the discussion about feasibility of this technique on a large-, commercial-scale. I wish I could understand your questions correctly and have given right answers in the revised manuscript. 

Bellow is the supplement discussion:

In modern commercial poultry farming systems, the production of broiler feed contributes up to 70% of the total production cost. Due to increases in global feed prices, there is now a tendency in the poultry industry to developing new high-quality protein source which can maximize animals growth performance while maintain good health[30]. FRSM is an ideal substitute feed ingredients for soybean in commercial poultry production. Owing to its low price, there is an increasing interest in incorporating FRSM into broiler rations to take advantages of its positive influences, particularly on production parameters and gut health.

Round 2

Reviewer 1 Report

The manuscript has been improved but there are still several items which raised my concern. Please see attached manuscript.

Abbreviations

Tables are not renumbered

Endogenous aminoa cid losses included but not discussed at all

References  used do not fit the statement given (especially  polypetides section discussion)

Especially the discussion is very poor, there is no clear mode of action described why digestibility or AME is increased.

Author Response

Thanks very much for your kind work and consideration on publication of our manuscript entitled “Effects of fermentation on standardized ileal digestibility of amino acids and apparent metabolizable energy in rapeseed meal fed to broiler chickens” (Manuscript ID animals-876031). Those comments are all valuable and very helpful for revising and improving our paper, as well as the important guiding significance to our researches. We have revised the manuscript according to your comments and suggestions, and the amendments are marked with red in the revised manuscript. Below you will find our point-by-point responses to your comments and questions. The whole manuscript has been carefully checked again by co-author  Shoaib Ahmed Pirzado.

We do hope we could understand your questions correctly and have given right answers in the revised manuscript. Please feel free to inform me if there are still any questions. Thank you very much!

Reviewer #1:

Question 1. the manuscript has been improved but there are still several items which raised my concern. Please see attached manuscript.

Abbreviations

Tables are not renumbered

Endogenous aminoa cid losses included but not discussed at all

References used do not fit the statement given (especially  polypetides section discussion)

Especially the discussion is very poor, there is no clear mode of action described why digestibility or AME is increased.

Answer: Thank you very much for your precious encouragement and suggestions. We totally agree with you and we re-prepared this manuscript carefully. We are so sorry to describe the discussion part unclear. We have revised the manuscript according to your comments and suggestions, and the amendments are singed in the revised manuscript.

Abbreviationswere added and Tabs were renumbered in the new manuscript.  

Endogenous amino acid losses were used to calculate the SID, it doesn’t make difference between RSM and FRSM. Similar way was published before, please see reference “Ullah Z.; Rehman Z.U.; Yin Y.; Stein H.H.; Sarwar M. Comparative ileal digestibility of amino acids in 00-rapeseed meal and rapeseed meal fed to growing male broilers. Poult. Sci. 2017, 96, 2736-2742”.

References were reordered in the revised manuscript.

The discussion part was reorganize, please see the revised manuscript.

Below you will find our point-by-point responses to your comments and questions. Furthermore, the words, fluency and logic were consider again carefully.

Question 2. P1 Simple Summary Again, why well balanced for poultry and you mean amino acid composition.

Answer: Thank you very much for your comment! We changed the description as “abundant sulfur-containing amino acids” in the revised manuscript.

Question 3. P1 Abstract  There are still abbreviations which are not introduced before first use.

Why do it in the line below...but not here

Answer: Thank you very much for your priceless question. We deleted the abbreviations in the revised manuscript.

Question 4. P1 Introduction  wrong spell and grammar about  because, lose ”

Answer: Thanks very much for your priceless question! We changed the spell in the revised manuscript followed your suggestion.

Question 5.  P2 Introduction  fermentation and drying cost as well and as can be seen from the results the increase in nutrive value is below 10 %... I have doubts that it will reduce the cost, but it may increase the use of rapeseed meal

Answer: Thank you very much for your priceless question. I want to express that fermentation can increase the use of rapeseed meal in formula, thereby it can reduce the cost of broiler production.

Question 6. P3Material and methods please give a reference for the EU standards which you follow

Answer: Thank you very much. We added the reference in the revised manuscript.

Question 7. P3 Material and methods  Please write full sentences

Answer: Thank you very much for your priceless suggestion. We changed the description.

Question 8. P2 Material and methodsare building, equipment and drying cost, personal cost here included .

Please calculate in US Dollar

Answer: Thank you very much for this precious suggestion. Yes, it is total cost ,and we calculated in us dollar in the new manuscript.

Question 9. P3 Material and methods give a reference for this guide!!

Answer: Thank you very much for your priceless suggestion. We added a reference about Arbor Acre Broiler management guide.in the revised manuscript.

Question 10.  P5 Material and methods Repeated statement...rewrite..see begining of 2.3.

Answer: Thanks very much. Here we mainly want to show the trial process, we made a simple change in the revised manuscript.

Question 11. P5 Material and methods Were is the feed intake data. How much feed did birds receive. This should be given somewhere in the text.

Answer: Thanks very much for your question. This experiment was designed to use a TiO2 marker method, the experiment diets were specially formulated, especially the NFD ( nitrogen-free diet) was nearly no protein. Besides, the official test just lasted few days, so, I think the feed intake and body weight gain didn’t means so much for the experiment. Some published paper also contains no data about feed intake or body weight gain(Nitrogen-corrected apparent metabolizable energy values of barley varies by treatment and species). Hope you understand why I didn’t provide feed intake data.

Question 12. P6 Material and methods adds up to 100,28 !!!

Answer:Thank you very much for your priceless question.we checked the formula and revised it, please refer to the new manuscript.

Question 13. P7 Material and methods This is an analysis for specific soypeptides...

This polypeptides is no measure of protein degradation as it measures only specific and not all polypeptides

Anyway ususally ammonia content or biogenic amines are measured for protein degradation.

Answer: Thank you very much for your priceless suggestion. The increased polypeptide of FRSM suggest an increase in the degree of protein hydrolysis, hence increasing the proportion of soluble low-molecular-size proteins, thereby making the protein more available for uptake by the chickens. We added more discussion about polypeptide index in the new manuscript.

Question 14.  P7 Material and methods are the concentration of the marker in the excreta and the diet

Answer: Thank you very much for this precious question. Yes, they are the concentration of the marker in the excreta and the diet.

Question 15. P9 results Tables have not been renumbered in the text..should be Table 8

Table 7 is not mentioned at all.

Answer: Thank you very much for your priceless suggestion. We renumbered the table, please refer the revised manuscript.

Question 16. P11 results We do not need both...the SEM and the standard deviation...one is sufficient !!!

Answer: Thanks very much for your priceless suggestion. We deleted the group SEM in the new manuscript.

Question 17. P12 Discussion This part is completely nonsense! To be honest I don't like to be fooled.

  1. The topic of Ref 31 are polyphenols no polypeptides measured.
  2. The Ref 32 no polypeptides measured.
  3. Ref 33 no polypeptides mentioned biogenic amines and losses of amino acids
  4. Ref 34 no polypeptides measured

I do not agree with this paragraph at all

Answer: Thanks very much for your priceless suggestion. The reference are update, please refer the advised manuscript.

Question 18. Discussion

Answer: Many thanks for your and question and suggestion. We revised the discussion part follow your advise, please refer the new manuscript.

Reviewer 2 Report

After reading the revised version, I can say that the Authors satisfactorily addressed the main concerns from the review.  However, some corrections are still required.

Introduction – please also correct abbreviations for Ala, Val, Ile, Leu, Tyr, Lys, Arg, and Phe with full names (also write them without capitalized letters)

In experiment 1 (in which a NFD diet was not used) average daily feed intake for each treatment was recorded. Therefore I think that for this experiment results some performance results can/should be provided/discussed.  

Author Response

Thanks very much for your kind work and consideration on publication of our manuscript entitled “Effects of fermentation on standardized ileal digestibility of amino acids and apparent metabolizable energy in rapeseed meal fed to broiler chickens” (Manuscript ID animals-876031). Those comments are all valuable and very helpful for revising and improving our paper, as well as the important guiding significance to our researches. We have revised the manuscript according to your comments and suggestions, and the amendments are marked with red in the revised manuscript. Below you will find our point-by-point responses to your comments and questions. The whole manuscript has been carefully checked again by co-author  Shoaib Ahmed Pirzado.

We do hope we could understand your questions correctly and have given right answers in the revised manuscript. Please feel free to inform me if there are still any questions. Thank you very much!

Reviewer #1:

Question 1. After reading the revised version, I can say that the Authors satisfactorily addressed the main concerns from the review.  However, some corrections are still required.

Introduction – please also correct abbreviations for Ala, Val, Ile, Leu, Tyr, Lys, Arg, and Phe with full names (also write them without capitalized letters)

In experiment 1 (in which a NFD diet was not used) average daily feed intake for each treatment was recorded. Therefore I think that for this experiment results some performance results can/should be provided/discussed.  

Answer: Thank you very much for your precious encouragement and suggestions. We totally agree with you and we re-prepared this manuscript carefully. We are so sorry to describe the introduction part unclear. We have revised the manuscript according to your comments and suggestions, and the amendments are singed in the revised manuscript.

We replaced the abbreviations with its full name in the revised manuscript.

This experiment was designed to use a TiO2 marker method, the experiment diets were specially formulated. Besides, the official test just lasted few days, so, I think the feed intake and body weight gain didn’t means so much for the experiment. Some published paper also contains no data about feed intake or body weight gain (Nitrogen-corrected apparent metabolizable energy values of barley varies by treatment and species). we designed another experiment specially to explore the effect of fermented rapeseed meal on the growth performance and immune status in broilers.

Hope you can understand why I didn’t provide feed intake data.

Besides, the discussion part was rewrite in the new manuscript.